# Trajectory Planning and Simulation Study of Redundant Robotic Arm for Upper Limb Rehabilitation Based on Back Propagation Neural Network and Genetic Algorithm

**DOI:** 10.3390/s22114071

**Published:** 2022-05-27

**Authors:** Xiaohan Qie, Cunfeng Kang, Guanchen Zong, Shujun Chen

**Affiliations:** Department of Materials and Manufacturing, Beijing University of Technology, Beijing 100124, China; qiexiaohan@emails.bjut.edu.cn (X.Q.); zongguanchen@emails.bjut.edu.cn (G.Z.); sjchen@bjut.edu.cn (S.C.)

**Keywords:** upper limb rehabilitation robotic arm, back propagation neural network, genetic algorithm, trajectory planning

## Abstract

In this study, a Back Propagation (BP) neural network algorithm based on Genetic Algorithm (GA) optimization is proposed to plan and optimize the trajectory of a redundant robotic arm for the upper limb rehabilitation of patients. The feasibility of the trajectory was verified by numerical simulations. First, the collected dataset was used to train the BP neural network optimized by the GA. Subsequently, the critical points designated by the rehabilitation physician for the upper limb rehabilitation were used as interpolation points for cubic B−spline interpolation to plan the motion trajectory. The GA optimized the planned trajectory with the goal of time minimization, and the feasibility of the optimized trajectory was analyzed with MATLAB simulations. The planned trajectory was smooth and continuous. There was no abrupt change in location or speed. Finally, simulations revealed that the optimized trajectory reduced the motion time and increased the motion speed between two adjacent critical points which improved the rehabilitation effect and can be applied to patients with different needs, which has high application value.

## 1. Introduction

Multi−Degree−Of−Freedom (MDOF) robotic arms and automation systems have been widely used in the field of welding and the tools used for welding were often installed on the end of the robotic arm, to replace the worker to complete some complex welding process. The welding action of the robotic arm and the corresponding various parameters have a great impact on the welding effect of the robotic arm. Kang et al. [1] proposed a control algorithm for weaving welding of circular trajectories based on the principle of spatial transformation, which can effectively solve the problem of discontinuities at the sharp corners of the weld, and found that the strategy can generate more accurate circular trajectories. Kang et al. [2] optimized the important parameters of the high−frequency welding process through finite element simulation. In addition, an automated chemical synthesis system was developed to perform chemical experiments [3]. However, the application of robotic arms in the biomedical field has only recently emerged. Bodner et al. [4] experimentally evaluated the suitability of the da Vinci surgical robot for thoracic surgery, proving that it was possible to safely perform thoracic surgery.

There are many factors that can cause damage to the upper limb. For example, stroke [5], prolonged smartphone use [6], and sports−related injuries [7]. Our upper limb rehabilitation robotic arm is not only for a specific type of disease, but also for all patients who need upper limb rehabilitation. Stroke is only used as a typical case to be illustrated in this study. Patients who have had a stroke are difficult to cure in a short time, and most of them require continuous care for rehabilitation [8]. This places a high demand on the number of health care workers. However, most countries have limited medical personnel to meet the needs of a large number of patients with disabilities. In response to the lack of medical staff, a wearable point−of−care system was proposed and researched to the continuously monitor patients’ vital signs and implements round−the−clock treatment protocols [9]. As for the rehabilitation of stroke patients, studies show that upper extremity motor restoration treatments, such as constraint−induced movement therapy and robotics, was effective in improving patients’ conditions [10].

All of the above causes can lead to loss of partial function of the upper limb in normal people. This will limit the movement of the upper limb to some extent, and in severe cases, the upper limb will not even be able to move on its own. As a result, there is a need to develop upper limb rehabilitation robotic arms to assist physicians in the rehabilitation of patients’ upper limb. Richardson et al. [11] developed a 3DOF pneumatic robot for physical therapy of the upper limb, and the device was experimentally shown to have great applicability. Johnson et al. [12] developed a 5DOF system for patient upper limb orthopedics with three control modes to perform different modes of orthopedic rehabilitation on the patient. Gao et al. [13] designed a virtual reality−based upper limb rehabilitation robot that avoided the problem of the large size of traditional MDOF rehabilitation robots and used a gyroscope to control a lightweight MDOF exoskeleton. However, the above−mentioned robotic arms with less than 6DOF all have a low flexibility of movement, which makes it difficult to meet the requirements in avoiding obstacles, or scenarios with special requirements for robot joint angles. It also fails to meet the requirements of different patients for rehabilitation. Compared to other lower degree−of−freedom robotic arms, our 7DOF upper limb rehabilitation robotic arm has more degrees of freedom to treat patients with different levels and causes of injury to adopt more positions for rehabilitation. Therefore, in this study, a 7DOF redundant robotic arm with a greater motion flexibility was selected for upper limb rehabilitation, which helped different patients to perform rehabilitation training in a more flexible way.

In the present, the application of machine learning and deep learning has become more and more widespread. For engineering applications, a hybrid gradient boosting method combining standard statistical and machine learning models was performed to predict engineering failures [14], and fatigue damage of turbine blades was assessed by a deep learning regression hierarchy strategy [15]. On the control side, tuning of the proportional integral derivative controller was performed by a machine learning search method [16], and the proportional integral derivative controller was optimized by the Genetic Algorithm (GA) [17]. Furthermore, there are extensive opportunities for machine learning−assisted physical therapy in the future [18].

However, solving the inverse kinematic solution of a MDOF robotic arm by the conventional methods was complicated but not highly accurate. Therefore, the machine learning algorithms were used to solve the inverse kinematic solution of the robotic arm. Tejomurtula et al. [19] proposed a solution based on a structured neural network that allows fast training of the network to obtain results. However, the accuracy of the method was still not high enough when applied to certain precision tasks. Nearchou [20] proposed a modified GA to optimize the end−effector position using a GA to achieve simultaneous minimization of end−effector position error and robot joint displacement. However, this method also has some limitations, such as poor reliability of the results and long computation time when calculating the complex problems. Köker [21] designed a method for redundant robotic arms that can improve the computational accuracy by optimizing the results of the floating−point part of the neural network training using a GA, and found that the error of the neural network was much reduced by the GA. In the present study, the “Köker’s idea” was used to calculate the inverse kinematic solution of a redundant robotic arm for the upper limb rehabilitation using a neural network algorithm optimized by the GA, which improved the accuracy of calculation results and avoided the disadvantages of GA and neural network to some extent.

When the robotic arm is in motion, the robotic arm must move along a predetermined trajectory and ensure smoothness and continuity of the motion at the robot’s end. Therefore, it is necessary to ensure that the location or speed of the joints did not change abruptly, which eliminated the large errors in the robotic arm and the patients were prevented from receiving additional injuries. That is why we need to perform trajectory planning [22]. Tian et al. [23] proposed a GA to search for effective and optimal solutions in the task trajectory. Gasparetto et al. [24] used a fifth−order B−spline curve for motion trajectory planning and verified the effectiveness of the method via simulation. Sometimes the planned trajectory does not meet the intended requirements, which requires optimization of the trajectory to meet the requirements. Ahuactzin et al. [25] used a GA to optimize the planned trajectory. Therefore, in this study, we planned the rehabilitation trajectory of the robotic arm with the cubic B−spline interpolation method and optimized the planned trajectory by the GA.

To ensure the accuracy and smoothness of the rehabilitation process of the upper limb, the eight critical points in the rehabilitation process were collected from rehabilitation physicians to plan the motion trajectory of the end of the 7DOF redundant upper limb rehabilitation of the robotic arm. To improve the accuracy of the inverse kinematic solution of the robotic arm, as well as the trajectory planning, an algorithm of Back Propagation (BP) neural network optimized by GA was used to calculate the inverse kinematic solution of the robotic arm. The results were used for trajectory planning with the cubic B−spline interpolation method and the trajectory was optimized on the basis of GA for time minimization, which increased the speed of motion between two adjacent critical points and improved the effectiveness of rehabilitation.

## 2. Back Propagation Neural Network Algorithm Based on Genetic Algorithm Optimization

The eight critical points were selected by the rehabilitator for the patient’s rehabilitation process in the same two−dimensional plane. The motion trajectory consisting of these eight critical points did not guarantee the continuity of speed at the end of the rehabilitation robotic arm, so the corresponding trajectory planning was required to ensure its rehabilitation effect.

In this section, the iterative algorithm of the GA optimized BP neural network was used to find the inverse kinematic solution of the robotic arm. Figure 1 shows the sketch of the robotic arm model. The BP neural network algorithm has the advantages of simple structure and easy implementation. However, it is time−consuming training, easy to fall into local optimum, and no ideal regulation to follow in the selection of the number of nodes in the hidden layer, which makes it impossible to determine the appropriate network structure. Therefore, the weights and thresholds of the neural network was iteratively calculating by a GA. The calculated results were then substituted into the neural network, and the collected dataset was used to train the neural network to acquire the optimal neural network. Finally, the above problem was solved using the optimal network for inverse kinematic computation.

We have fabricated and built a redundant robotic arm for upper limb rehabilitation and have performed some test works on it. Figure 2 shows the SolidWorks model and physical object. The following studies are all based on the upper limb rehabilitation of the 7DOF redundant robotic arm.

### 2.1. Principles of Back Propagation Neural Network and Genetic Algorithm

#### 2.1.1. Back Propagation Neural Network

The BP neural network is a multilayer feedforward neural network trained according to a back−transmission error algorithm, and Figure 3 schematically shows the structure of the neural network of the input layer, hidden layer, and output layer. Each layer contains many neurons inside, and the computational unit in the neuron is the activation function. The activation functions commonly used in network computation are the Sigmoid function [26] and Tansig function [27]. During the operation of the network, data are transmitted sequentially with the input layer, hidden layer, and output layer. When the data reach the output layer they are then transmitted back, thus correcting errors in the network. The network calculation process contains two types of parameters: weights and thresholds. The magnitude of the values affects the error of the network.

The network contains two stages in the calculation, the phase of forwardly transmission of the signal and backwardly transmission of the error. Before training the neural network, the target value of the final output result should be set first. In the process of training the neural network, the size of the weights and thresholds are constantly revised, and the actual value of the output is constantly close to the preset target value, and the optimal neural network was finally obtained.

The first step is the web forward transmission. Let the nodes in the input layer of the network be numbered *j* (*j* = 1, 2, …, *v*), the nodes in the hidden layer of the network are assigned numbers *n* (*n* = 1, 2, …, *u*), and the nodes in the output layer of the network are given numbers *k* (*k* = 1, 2, …, *z*). The signal passes the data through the input layer to the hidden layer. The input *N_n_* of the *n*th node of the input layer is given by Equation (1):(1)$$N_{n}=\sum_{j=1}^{v}w_{nj}x_{j}+b_{n}$$
where *b_n_* is the threshold of the *n*th node in the hidden layer, *w_nj_* is the weight of the *j*th node in the input layer to the *n*th node in the hidden layer, and *x_j_* is the input value of the *j*th node in the input layer. Substituting Equation (1) into the activation function *f*_h_ in the hidden layer, the output of the *n*th node of the hidden layer is given as follows:(2)$$y_{n}=f_{h}\left(\sum_{j=1}^{v}w_{nj}x_{j}+b_{n}\right)$$

Using the output of the hidden layer as the input of the output layer, the input of the *k*th node in the output layer is obtained as follows:(3)$$N_{k}=\sum_{n=1}^{u}w_{kn}y_{n}+a_{k}$$
where *a_k_* is the threshold value of the *k*th node of the output layer, *w_kn_* is the weight of the *n*th node in the input layer to the *k*th node in the output layer, and *y_n_* is the output value of the *n*th node of the hidden layer. Finally, the output equation for the *k*th node of the output layer was obtained by substituting the Equation (3) into the activation function *f*_o_ of the output layer:(4)$$Q_{k}=f_{o}\left(N_{k}\right)$$

The second step is the network back transmission. When the forward transmission of the signal was completed, the network will calculate the output error of the neurons in each layer in reverse from the output layer. Then the error gradient descent method was used to adjust the weights and thresholds of each layer. Firstly, the Mean Square Error (MSE) *E_p_*, of a single sample datum *p* was calculated by the following equation:(5)$$E_{p}=\frac{1}{2}\sum_{k=1}^{z}{\left(T_{k}-Q_{k}\right)}^{2}$$
where *T_k_* is the expected value of the *k*th node. Using the gradient descent method to find the weight and threshold correction amount of the output layer and the hidden layer, respectively. The specific equations are shown in Equations (6)–(9), where ∆*w_kn_* and ∆*w_nj_* represent the weight correction of the output layer and the hidden layer, respectively, ∆*a_k_* and ∆*b_n_* denote the threshold correction of the output layer and the hidden layer, respectively.
(6)$$\Delta w_{kn}=-\eta \frac{\partial E}{\partial w_{kn}}=\eta \sum_{p=1}^{P}\sum_{k=1}^{z}\left(T_{k}^{p}-Q_{k}^{p}\right)\times f_{o}{}^{\prime }\left(N_{k}\right)\times y_{n}$$
(7)$$\Delta a_{k}=-\eta \frac{\partial E}{\partial a_{k}}=\eta \sum_{p=1}^{P}\sum_{k=1}^{z}\left(T_{k}^{p}-Q_{k}^{p}\right)\times f_{o}{}^{\prime }\left(N_{k}\right)$$
(8)$$\Delta w_{nj}=-\eta \frac{\partial E}{\partial w_{nj}}=\eta \sum_{p=1}^{P}\sum_{k=1}^{z}\left(T_{k}^{p}-Q_{k}^{p}\right)\times f_{o}{}^{\prime }\left(N_{k}\right)\times w_{kn}\times f_{h}{}^{\prime }\left(N_{n}\right)\times x_{j}$$
(9)$$\Delta b_{n}=-\eta \frac{\partial E}{\partial b_{n}}=\eta \sum_{p=1}^{P}\sum_{k=1}^{z}\left(T_{k}^{p}-Q_{k}^{p}\right)\times f_{o}{}^{\prime }\left(N_{k}\right)\times w_{kn}\times f_{h}{}^{\prime }\left(N_{n}\right)$$
where *η* is the learning rate of the network, and *P* is the number of training samples.

#### 2.1.2. Genetic Algorithm

GA is to generate a new population by continuously searching to select the optimal solution, the specific process is similar to the process of human genetic evolution. The specific process of GA is the selection and initialization of the population, fitness function, selection, crossover, mutation, getting new individuals, and judging whether the new individuals meet the required conditions. If they do not meet the conditions, then return to the selection operator to continue the search for the optimal solution. If they meet the conditions, then the optimal solution is the output. Individuals are often encoded by binary encoding, multiparameter cascade encoding, and floating−point encoding. The encoding methods have their characteristics and were adapted to different practical application scenarios. During the operation process, the individuals were selected and crossed between two operators. The individuals themselves are affected by the variation operators.

### 2.2. Computational Procedure of Back Propagation Neural Network Algorithm Based on Genetic Algorithm Optimization

In the present study, the process of the BP neural network algorithm based on GA optimization has four steps. Figure 4 shows the flow chart of the computation.

In the first step, a dataset was created, and each joint angle was divided equally into 180 parts in its range of motion. The corresponding end location matrices were solved using the forward kinematic equations, and these matrices were transformed to correspond joint matrices. The 180 datasets were generated and stored as the elements of the input and output layers in the neural network, respectively. In the second step, a three−layer BP neural network model was built. The number of input layer neurons *I* was set to 12, the number of output layer neurons *O* was set to 7, and the number of hidden layer neurons *H* was calculated according to the empirical Equation (10). The number of neurons in the hidden layer was calculated to be 14 and the Pureline function [28] was selected as the activation function for the input and output layers in this study. The training times of the network were set to 1000, the training target was set to 0.0001, and the learning rate was set to 0.1.
(10)$$H=\sqrt{I+O}+\alpha$$
where *α* is a regulation constant with a default value range of 1 to 10. The third step is to establish a GA model. This model used binary coding, the range of the weight threshold value was limited between −5 to 5, and the individual precision was set to 0.01. According to Equation (11), *L* was 9.967, and was rounded to 10, thus the individual binary coding length was 10. The primitive population was then determined, and the number of weights was equal to the number of neurons in the input layer multiplied by the number of neurons in the hidden layer, plus the number of neurons in the hidden layer multiplied by the number of neurons in the output layer. The number of thresholds was equal to the number of neurons in the hidden layer, plus the number of neurons in the output layer. The network contains 287 weights and thresholds, of which 266 were weights and 21 were thresholds on the basis of the number of neurons in the three layers of the neural network. Forty groups of individuals were randomly generated as the parents and a matrix of size 40 × 2870 was formed and then was substituted into the neural network. The errors calculated by the neural network after bringing in the matrix were sorted to create the fitness function. Finally, the individuals were updated iteratively by three operators of selection, crossover, and variation until the set number of genetic generations was completed. As seen in Figure 5, with continuous evolutionary inheritance, the two norm error decreases in steps, and finally, the weights and thresholds with the smallest errors are output.
(11)$$L=\log_{2}\left(\frac{b-a}{esp}+1\right)$$
where *L* is the number of binary codes, *a* is the lower bound for individual values, *b* is the upper bound on the value taken by an individual, and *esp* is the accuracy of the individual.

In the fourth step, the optimal weights and thresholds obtained by the GA were substituted into the built BP neural network model. The first 150 sets of the dataset were used for the training of the network, and the last 30 sets were used to verify the network training results. As seen in Figure 6, with the number of training sessions increasing, the MSE decreased continuously. The curve of the train and test set has almost the same trend, thus the network meets the requirement.

To verify the error of the trained network, the angles of the seven joints were divided into 500 parts and substituted into the trained network to analyze the error. The maximum error between the solved data and the original data is 0.002 rad. As shown in Table 1 and Table 2, 10 sets of angle information were selected in the table. One was the original radian dataset, and the other was the radian dataset solved by the trained network. By comparing the original radian data and the radian data, the error was very small. After verifying the training results of the network, the collected data of eight critical points were input to the trained network. The network outputs the angles of each joint, thus completing the calculation of the inverse kinematic solution of the 7DOF robotic arm

We found the following methods for solving the inverse kinematic solution of the robotic arm in other articles: Jacobian transpose [29], neural network [30], Multiple Population Genetic Algorithm (MPGA) [31], Firefly Algorithm (FA), and Artificial Bee Colonies (ABC) [32]. The results of the comparison between these methods and the methods in this study are shown in Table 3. The comparison results in the table showed that the method used in this study had a higher accuracy than the other methods of calculation.

## 3. Trajectory Planning of Redundant Robotic Arm for Upper Limb Rehabilitation

In this section, the rehabilitation curve was planned by the cubic B−spline interpolation method. The B−spline curve has the qualities of local support, composite and convex envelope compared with other curves, which can quickly obtain the calculation results. Thus, the cubic B−spline interpolation method was used to plan the rehabilitation trajectory. The inverse kinematic solution of the eight critical points was solved by the above GA−optimized BP neural network algorithm. The critical points were planned into a complete rehabilitation curve by B−spline interpolation in the Cartesian coordinate system, and the kinematic analysis of each joint was performed in the joint space to verify the smoothness of the trajectory motion.

### 3.1. Derivation of Cubic B−Spline Interpolation

During trajectory planning using the cubic B−spline interpolation method, the set of position points passed by each joint was known, and the planning of robotic arm trajectory was realized by back−calculating the control vertices of the B−spline curve to find the angle variables of each joint as a function of time.

The equation of the cubic B−spline curve function for each joint of the robot arm with the angle variable *θ* at time *t* is shown as follows:(12)$$\begin{array}{ccc} \theta_{i}\left(t\right) & = & A_{0}\left(t\right)V_{i-1}+A_{1}\left(t\right)V_{i}+A_{2}\left(t\right)V_{i+1}+A_{3}\left(t\right)V_{i+2} \end{array}$$

The first−order and second−order derivatives of Equation (12) could be obtained for the joint angular speed and joint angular acceleration for the time in Equation (13), where the value of time *t* ranges from 0 to 1, *A* is the basis function, *V* is the control vertex. The variable *i* (*i =* 1, 2, …, *m*) indicated the *i*th section of the curve, whose corresponding control vertices were numbered *V_i_*_−1_, *V_i_*, *V_i_*_+1_, *V_i_*_+2_, and four adjacent control vertices constituted a set of control points.
(13)$$\{\begin{array}{c} {\dot{\theta }}_{i}\left(t\right)={\dot{A}}_{0}\left(t\right)V_{i-1}+{\dot{A}}_{1}\left(t\right)V_{i}+{\dot{A}}_{2}\left(t\right)V_{i+1}+{\dot{A}}_{3}\left(t\right)V_{i+2}\\ {\ddot{\theta }}_{i}\left(t\right)={\ddot{A}}_{0}\left(t\right)V_{i-1}+{\ddot{A}}_{1}\left(t\right)V_{i}+{\ddot{A}}_{2}\left(t\right)V_{i+1}+{\ddot{A}}_{3}\left(t\right)V_{i+2} \end{array}$$

From the previous sections, the B−spline curve has continuity, and the end of the *i*th segment of the curve was continuous with the start of the (*i* + 1)th segment of the curve, which yields Equation (14):(14)$$\theta_{i}\left(1\right)=\theta_{i+1}\left(0\right)$$

Substituting Equation (14) into Equation (12), Equation (15) was obtained:(15)$$A_{0}\left(1\right)=A_{3}\left(0\right)=0,\;A_{1}\left(1\right)=A_{0}\left(0\right),\;A_{2}\left(1\right)=A_{1}\left(0\right),\;A_{3}\left(1\right)=A_{2}\left(0\right)$$

From the continuity of the first−order and second−order derivatives of the B−spline curve, the following equation could be derived:(16)$${\dot{\theta }}_{i}\left(1\right)={\dot{\theta }}_{i+1}\left(0\right),{\ddot{\theta }}_{i}\left(1\right)={\ddot{\theta }}_{i+1}\left(0\right)$$

Substituting Equation (16) into Equation (13) yielded the following results:(17)$$\{\begin{array}{c} {\dot{A}}_{0}\left(1\right)={\dot{A}}_{3}\left(0\right)=0,{\dot{A}}_{1}\left(1\right)={\dot{A}}_{0}\left(0\right),{\dot{A}}_{2}\left(1\right)={\dot{A}}_{1}\left(0\right),{\dot{A}}_{3}\left(1\right)={\dot{A}}_{2}\left(0\right)\\ {\ddot{A}}_{0}\left(1\right)={\ddot{A}}_{3}\left(0\right)=0,{\ddot{A}}_{1}\left(1\right)={\ddot{A}}_{0}\left(0\right),{\ddot{A}}_{2}\left(1\right)={\ddot{A}}_{1}\left(0\right),{\ddot{A}}_{3}\left(1\right)={\ddot{A}}_{2}\left(0\right) \end{array}$$

Furthermore, according to the normality of the B−spline curve, the following equation could be obtained:(18)$$A_{0}\left(t\right)+A_{1}\left(t\right)+A_{2}\left(t\right)+A_{3}\left(t\right)=1$$

In this study, the cubic B−spline curve interpolation method was used, so the equation basis function was set as follows, where *a_i_*, *b_i_*, *c_i_*, and *d_i_* are the four coefficients assumed:(19)$$A_{i}\left(t\right)=a_{i}+b_{i}t+c_{i}t^{2}+d_{i}t^{3}$$

Combining Equations (12) to (19), an expression for the cubic B−spline curve equation for the joint angle at a given time in the joint space of the robot arm was obtained:(20)$$\theta_{i}\left(t\right)=\frac{1}{6}(1-3t+3t^{2}-t^{3})V_{i+1}+\frac{1}{6}t^{3}V_{i+2}+\frac{1}{6}(4+3t^{3}-6t^{2})V_{i}+\frac{1}{6}(1+3t+3t^{2}-3t^{3})V_{i+1}$$

By solving the first and second−order derivatives of Equation (20), the equations of angular velocity and angular acceleration was obtained as follows:(21)$${\dot{\theta }}_{i}\left(t\right)=-\frac{1}{2}{\left(1-t\right)}^{2}V_{i-1}+\frac{1}{2}\left(3t^{3}-4t\right)V_{i}+\frac{1}{2}\left(-3t^{2}+2t+1\right)V_{i+1}+\frac{1}{2}t^{2}V_{i+2}$$
(22)$${\ddot{\theta }}_{i}\left(t\right)=\left(1-t\right)V_{i-1}+\left(3t-2\right)V_{i}+\frac{1}{2}\left(-3t+1\right)V_{i+1}+tV_{i+2}$$

To clarify the relation between the variables of Equation (20), it was rewritten in the form of a matrix with the following expression:(23)$$\theta_{i}\left(t\right)=\frac{1}{6}\left[{\begin{array}{cccc} 1 & t & t^{2} & t \end{array}}^{3}\right]\left[\begin{array}{cccc} 1 & 4 & 1 & 0\\ -3 & 0 & 3 & 0\\ 3 & -6 & 3 & 0\\ -1 & 3 & -3 & 1 \end{array}\right]\left[\begin{array}{c} V_{i-1}\\ V_{i}\\ V_{i+1}\\ V_{i+2} \end{array}\right]$$

The value of the control vertex *V_i_* was unknown, and the control vertex was found by inputting the joint coordinates of the robotic arm. The points on the trajectory of the actuators were known, and the joint angle was obtained by the inverse kinematics. Taking joint #1 as an example, the curve passed through points *P*_1_, *P*_2_, …, *P_m_*, and the two adjacent points were planned by a section of the cubic B−spline curve. A total of *m* − 1 segments of the B−spline curve existed. In addition, because of the continuity of the curve, Equation (24) could be obtained:(24)$$\theta_{i+1}\left(1\right)=\theta_{i}\left(0\right)=\frac{1}{6}V_{i-1}+\frac{2}{3}V_{i}+\frac{1}{6}V_{i+1}=P_{i}$$

The equation containing *m* data points described the geometry of the curve or surface and *m* + 2 control point vertices. To ensure the smoothness and continuity of the curve, two constraints were added as *V*_0_ = *V_m_
*and *V*_1_ = *V_m_*_+1_. Thus Equation (24) was transformed into the form of a matrix:(25)$$\left[\begin{array}{ccccccc} 1 & 4 & 1 & & & & 1\\ & 1 & 4 & 1 & & & \\ & & & \ldots & & & \\ & & & & 1 & 4 & 1\\ 1 & & & & & 1 & 4\\ 4 & 1 & & & & & 1 \end{array}\right]\left[\begin{array}{c} V_{0}\\ V_{1}\\ \ldots \\ V_{m-3}\\ V_{m-2}\\ V_{m-1} \end{array}\right]=6\left[\begin{array}{c} P_{1}\\ P_{2}\\ \ldots \\ P_{m-2}\\ P_{m-1}\\ P_{m} \end{array}\right]$$

The system of equations had *m* equations and unknown quantities, which was solved for a unique set of solutions. The values of the control vertices *V*_0_ to *V_m_*_−1_ was obtained by this solution set. Subsequently, all the control vertices *V*_0_ to *V_m_*_+1_ were derived by the two constraints. Thus, the solution was completed for each section of the trajectory B−spline curve equation.

### 3.2. Rehabilitation Trajectory Planning for Redundant Robotic Arm for Upper Limb Rehabilitation

#### 3.2.1. Rehabilitation Trajectory Planning under Cartesian Space

Under the Cartesian space, the coordinates of the eight critical points, which were solved by the inverse kinematic solution in Section 2, were transformed to the coordinates under the Cartesian coordinate system by the forward kinematic solution. The trajectories of the eight critical points after the forward kinematic solution were planned using the cubic B−spline interpolation method derived above to form a complete section of the spatial curve, as shown in Figure 7. The green line in the figure is the original rehabilitation trajectory. The blue line is the optimized trajectory with an increasing movement range. For people with upper limb disorders, there are many unreachable spatial locations. In this study, we assumed that the patient had a disability in the left upper limb, and the left side of the patient corresponds to the negative direction of the *Y*−axis in the figure. However, after a period of upper limb rehabilitation, the patient’s reachable range of the upper limb increased. Therefore, the motion trajectory of the robotic arm needs to expand accordingly, which is the reason why the trajectory optimization was needed.

#### 3.2.2. Joint Kinematic Analysis of Redundant Robotic Arm for Upper Limb Rehabilitation

Based on the trajectory planned with the cubic B−spline interpolation method under the Cartesian space above, the kinematic analysis was performed for each joint under the joint space to verify the feasibility of the trajectory. Take joint #1 as an example, its angles at each of the pre−given critical points during the motion were: *q* = [81.40°, 118.09°, 128.92°, 117.92°, 82.55°, 42.88°, 38.98°, 42.42°]. Set the corresponding time interval according to the angle difference between adjacent points, specific time intervals were: *t* = [8.0 s, 2.5 s, 2.5 s, 4.0 s, 4.0 s, 2.0 s, 2.0 s] and it took a total of 25 s to complete the rehabilitation trajectory. As shown in Figure 8, the circles in the location curve represent the eight critical points, and the cubic B−spline interpolation method is used to connect the eight critical points. We wrote an algorithmic program based on MATLAB to calculate the speed and acceleration of the joints. For the acceleration calculation, we only calculated the acceleration at several locations during the motion and connected them with straight lines. In this study, we mainly focus on the continuity of the speed curve and the curves passed all the preset points and the cubic B−spline interpolation method was able to adapt to the complex joint motion and completed the planning of the rehabilitation trajectory. The speed was maintained smoothly during the motion, which verified the stability of the rehabilitation robotic arm motion.

In addition to interpolation method of the cubic B−spline curve, we also used the cubic polynomial interpolation, the five−polynomial interpolation, and the seven−polynomial interpolation method to plan the motion trajectory of the robotic arm. In addition, we performed trajectory planning as well as MATLAB simulation for each method following the procedure of cubic B−spline interpolation.

First is the cubic polynomial interpolation method, the simulation results are shown in Figure 9.

The second is the five−polynomial interpolation method, the simulation results are shown in Figure 10.

Finally, the simulation results of the seven−polynomial interpolation are shown in Figure 11.

The simulation results show that the location and speed of the trajectories planned by these three methods were relatively stable and ensured the smoothness of the motion of the robotic arm.

## 4. Optimization of Rehabilitation Trajectory of Redundant Robotic Arm for Upper Limb Rehabilitation and Its Simulation Experiment

### 4.1. Optimization of Rehabilitation Trajectories with the Goal of Time Minimization

Since different patients have different needs for the upper limb rehabilitation, we need to expand the applicable range of the redundant robotic arm for upper limb rehabilitation. We can change certain parameters in the rehabilitation process using trajectory optimization to achieve the purpose of expanding the applicable range. For example, we could change the motion time of the end of the robotic arm between the critical points and thus change the speed of the end movement. We could also modify the location of the critical points and change the shape of the rehabilitation trajectory to meet the needs of different patients for rehabilitation. The curve after trajectory planning consists of multiple cubic B−spline curves that have been connected. Taking the total running time of the robot arm as the optimization objective, the running time of each segment of the curve was set to *t_i_*. The time of each part of the trajectory was optimized separately to obtain the final result. The objective function was set as follows:(26)$$T=\min\sum_{i=1}^{m-1}t_{i}$$
where *T* is the total operating time of the robotic arm. For time−targeted optimization, the motion time could not be reduced indefinitely and constraints needed to be added. Kinematic constraints include speed, acceleration, and jerk (derivative of acceleration).

First is the speed constraint. The joint angular speed equation was derived from the joint equation to the first−order derivative of time, the maximum angular speed of the robotic arm in motion might be at any point on the interval when the robotic arm motion time was at *t_i_
*to *t_i_*_+1_, there was an expression as follows:(27)$$\dot{\theta }=\max\left\{\left|{\dot{\theta }}_{i}\right|,\left|{\dot{\theta }}_{ix}\right|\right\}$$
where $\dot{\theta }$ is the maximum angular speed, $\left|{\dot{\theta }}_{i}\right|$ is the absolute value of the angular speed at moment *t*_i_, and $\left|{\dot{\theta }}_{ix}\right|$ is the maximum angular speed in the interval *t_i_* to *t_i_*_+1_. It was difficult to solve the equation for the absolute maximum value, and the golden mean method was used to solve the equation for the maximum value of ${\dot{\theta }}_{\max}$. The constraint form was given as follows:(28)$$\dot{\theta }\leq {\dot{\theta }}_{\max}$$

The second is the acceleration constraint. The joint angular acceleration equation was obtained from the joint angle equation by taking the second−order derivative of time and solving for the maximum value in the same way as the speed constraint. The maximum joint angular acceleration equation was solved as follows:(29)$$\ddot{\theta }=\max\left\{\left|{\ddot{\theta }}_{i}\right|,\;\left|{\ddot{\theta }}_{ix}\right|\right\}$$
where $\ddot{\theta }$ is the maximum angular acceleration, $\left|{\ddot{\theta }}_{i}\right|$ is the absolute value of angular acceleration at moment *t_i_*, and $\left|{\ddot{\theta }}_{ix}\right|$ is maximum angular acceleration in absolute value in the interval *t_i_* to *t_i_*_+1_. The acceleration constraint form of the robotic arm was as follows:(30)$$\ddot{\theta }\leq {\ddot{\theta }}_{\max}$$

The third is the jerk constraint. The joint angular jerk equation was solved for the maximum joint angular acceleration equation by taking the third−order derivative of the joint angle equation for a time and was given as follows:(31)$$\dddot{\theta }=\max\left\{\left|{\dddot{\theta }}_{i}\right|,\;\left|{\dddot{\theta }}_{ix}\right|\right\}$$
where $\dddot{\theta }$ is the maximum angular jerk, $\left|{\dddot{\theta }}_{i}\right|$ is add the absolute value of angular jerk at moment *t_i_*, and $\left|{\dddot{\theta }}_{ix}\right|$ is the absolute maximum plus angular jerk in the interval *t_i_
*to *t_i_*_+1_.

### 4.2. MATLAB Numerical Simulation of Redundant Robotic Arm for Upper Limb Rehabilitation

We assumed that the model had *m* joint angle values and the motion time between each adjacent two points was *t_i_*, a GA was used to find the optimal solution for *t_i_* in the model. In this study, the time interval matrix between two adjacent points was set as *t* = [8.0 s, 2.5 s, 2.5 s, 4.0 s, 4.0 s, 2.0 s, 2.0 s, and 50 groups of individuals were randomly generated at each given time. For each random individual, the fitness size *f* was assigned by the following equation.
(32)$$f=\left\{ \begin{array}{ll} \frac{1}{t_{i}}, & \mathrm{When}\;\mathrm{the}\;\mathrm{constraint}\;\mathrm{was}\;\mathrm{satisfied}\\ \frac{1}{t_{i}{}^{\max}}, & \mathrm{Otherwise} \end{array}\right.$$

The final results were calculated by the iterating depending on the iteration rules of the GA. According to the application scenario of the upper limb rehabilitation, the maximum joint angular speed *V*_max_ in the algorithm was set to 15°/s, the maximum joint angular acceleration *A*_max_ was set to 20°/s^2^, and the maximum joint angular jerk *I*_max_ was set to 100°/s^3^. Under the condition of satisfying the kinematic constraints, the total motion time of the robotic arm before the optimization was 25 s, which was reduced to 20.44 s after the time optimization. The time comparison before and after track optimization is shown in Table 4.

We wrote the above algorithm based on MATLAB. The kinematic analysis was performed on the curve obtained by cubic B−spline interpolation for joint #1 of the robotic arm and the trajectory optimized by the GA. As shown in Figure 12, the red line indicates the unoptimized analysis results and the blue line represents the optimized results. The results show that after optimization, the location and angular speed profiles of the joints were smooth, and the rehabilitation movements were completed in a shorter period of time, which expanded the range of applicability. In terms of safety, the patient had a certain tolerance after a period of rehabilitation training. Only after the patient had received a period of formal rehabilitation training, the time was shortened and the rehabilitation effect was improved. In addition, even though the acceleration curve changed more than the unoptimized curve, this change was still in a controlled range to ensure that the patient can receive secondary injuries during the rehabilitation process.

We have read the literature to the extent of our ability and describe the structure and movement of the upper limb as follows [33].

In terms of the upper limb structure, the upper limb consists of the shoulder girdle and the arm. The shoulder girdle consists of the clavicle and the scapula. The shoulder joint, which connects the arm to the scapula, is a multiaxial synovial ball and socket joint. The humeral head was shaped as a ball which coincides with the spherical surface of the glenoid cavity of the scapula, and the surfaces of the humeral head and cavity were covered with articular cartilage. In addition, the joint capsule was strengthened by muscles, tendons, and ligaments such as the glenohumeral, coracohumeral, and coracoacromial ligaments.

The arm was built of the humerus, ulna, radius bone, 8 carpal bones of the wrist, 5 metacarpal bones, and 14 phalanges of the fingers. The elbow consists of the humeroulnar, humeroradial, and proximal radioulnar joints, which were surrounded by a common joint capsule. The frontal stability was mainly ensured by the ulnar collateral ligament and the radial collateral ligament. Patients with impaired upper limb function usually suffer from wear and tear of the ligaments or joints of the upper limb. Therefore, the upper limb rehabilitation robotic arm in this study treated the patient’s upper limb by helping the patient to perform upper limb movements through end−traction method of the robotic arm.

In terms of the upper limb movement, the upper limb skeletal movement was produced by the 43 muscles being connected to structures mentioned above. The shoulder joint allows for abduction/adduction, flexion/extension, and internal/external rotation. At the elbow joint, there is elbow flexion/extension and forearm supination/pronation. In the wrist, there is flexion/extension and radial/ulnar deviation. However, because of the presence of the upper limb joints, there is a limited range of motion in each of these joints. Therefore, the upper limb rehabilitation of the robotic arm in this study planned the rehabilitation trajectory by pre−given critical points, thus avoiding the rehabilitation trajectory to reach the position where the patient’s upper limb cannot reach, which can ensure the safety of the patient during the rehabilitation process.

This redundant robotic arm for upper limb rehabilitation is still in the stage of theoretical modeling, numerical simulation, development, and testing of the prototype. Therefore, for safety reasons, it has not been commissioned and applied to patients yet. However, to obtain a better perceptual feedback on the human body, we had conducted preliminary tests with the manufactured robotic arm on several patients with healthy upper limb. These patients, with their sensitive upper limb perception, were able to provide better feedback on the treatment effect of the upper limb rehabilitation of the robotic arm, the comfort level during treatment (no sudden changes in force) and the continuity of the rehabilitation trajectory (continuous and smooth location and speed of the robotic arm). The rehabilitation movement of the end traction was performed according to the planned trajectory in this study, as shown in Figure 13.

The rehabilitation trajectory passed the eight critical points mentioned in this study, and the positions and postures of the robotic arm and the treated person during the rehabilitation process corresponded to a to h in Figure 13. The feedback from the treated person could better reflect the rehabilitation effect of the robotic arm, so that the trajectory of the robotic arm could be adjusted and optimized accordingly before practical application. For example, the trajectory optimization method with the goal of time minimization in this study could change the speed during the motion and the magnitude of the force applied to the upper limb. Alternatively, the range and accuracy of the motion trajectory of the robotic arm end could be adjusted by changing the location of the critical points and adjusting the number of critical points. The test results show that the robotic arm did not cause excessive compression on the upper limb during the rehabilitation process guided by the end. The force and speed applied to the human upper limb were moderate, and the end of robotic arm was relatively smooth during the movement, which could achieve the effect of assisting in guiding the upper limb rehabilitation.

We will cooperate with the TCM rehabilitators from Beijing Red Medical Star Intelligent Technology Development Company Limited in the future and use the robotic arm in the actual rehabilitation process. We will discuss the effectiveness of the treatment and the practical application in the next articles.

## 5. Conclusions

In this study, we proposed a BP neural network algorithm optimized by GA. Compared with other algorithms, the computational results of this algorithm have better accuracy. We used cubic B−spline interpolation method for trajectory planning and performed numerical simulation with MATLAB. The results show that the location and speed profiles of the planned curve were relatively smooth and could meet the needs of patient rehabilitation. To further improve the rehabilitation effect, we optimized the planned trajectory based on GA with the goal of time minimization, and the numerical simulation results in MATLAB showed that the optimized curve reduced the motion time, thus increasing the motion speed of the end of the robotic arm and improving the rehabilitation effect. For safety reasons, we have not yet applied this upper limb rehabilitation of the robotic arm to patient rehabilitation treatment. We had tested it on several patients with normal upper limb and the results show that the force and speed applied to the upper limbs during the rehabilitation process was relatively stable and could achieve the effect of assisting in guiding the rehabilitation of the upper limbs. In the future, we will cooperate with TCM rehabilitators and use the upper limb rehabilitation of the robotic arm in the rehabilitation process of patients. We will discuss the treatment effect and practical application of the upper limb rehabilitation arm in our future work.

## Figures and Tables

**Figure 1 sensors-22-04071-f001:**
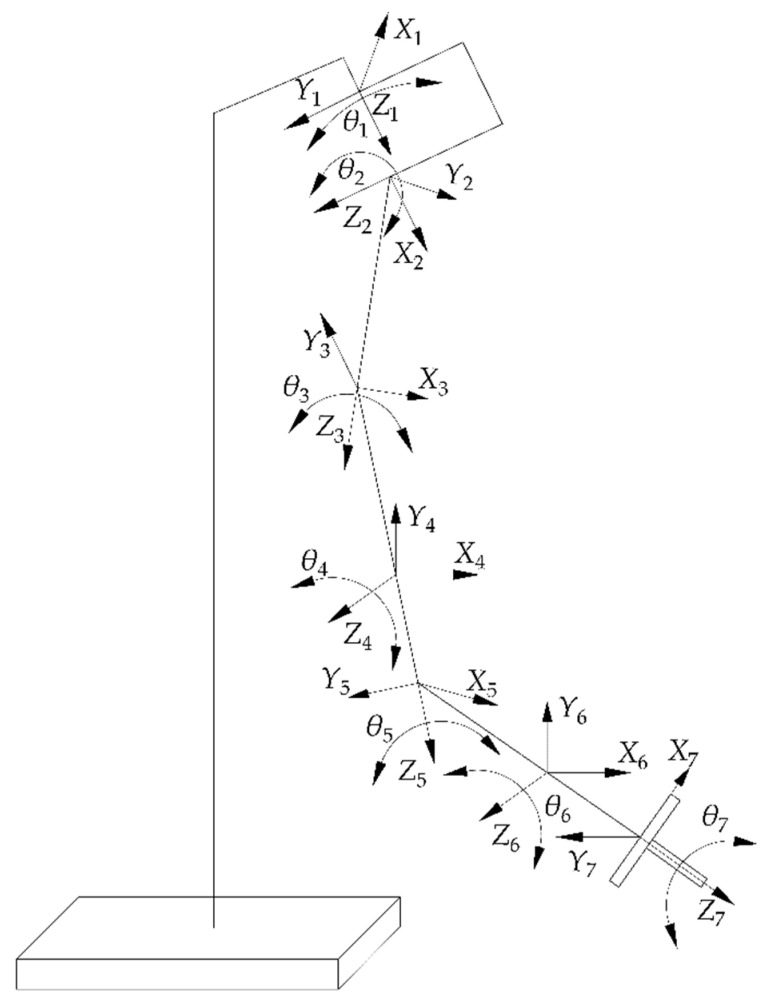
Sketch of a redundant robotic arm for 7 Degree−Of−Freedom upper limb rehabilitation.

**Figure 2 sensors-22-04071-f002:**
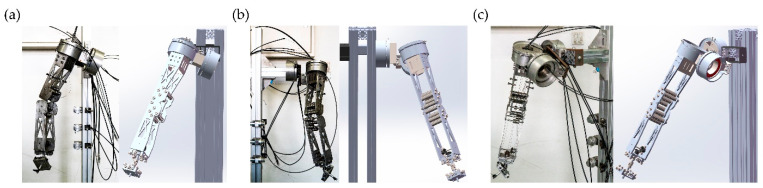
SolidWorks model and physical correspondence diagram. (**a**) Front; (**b**) Left; (**c**) Diagonal.

**Figure 3 sensors-22-04071-f003:**
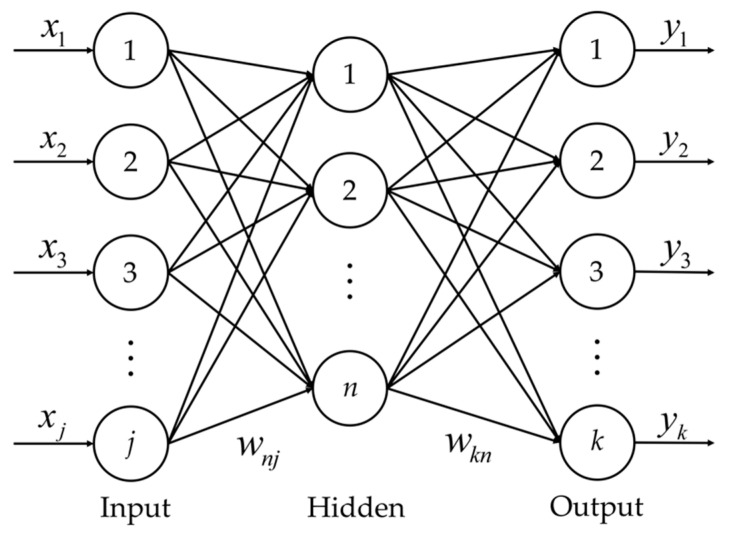
Schematic diagram of Back Propagation neural network structure.

**Figure 4 sensors-22-04071-f004:**
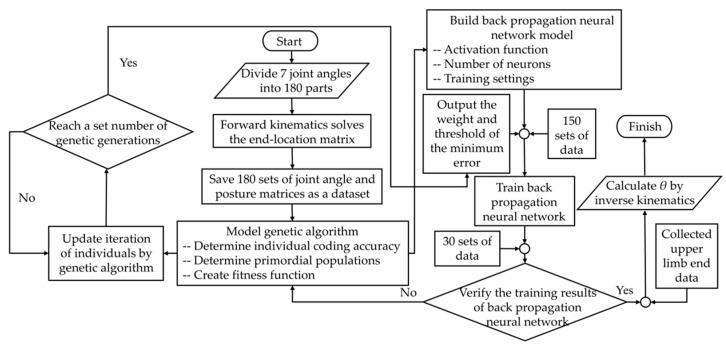
Computational flow chart.

**Figure 5 sensors-22-04071-f005:**
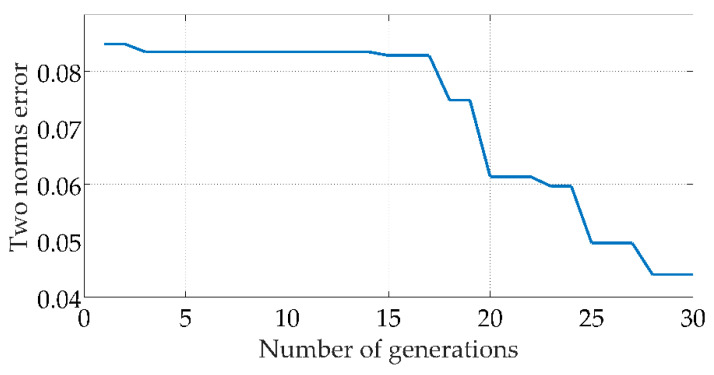
Two norms error curve.

**Figure 6 sensors-22-04071-f006:**
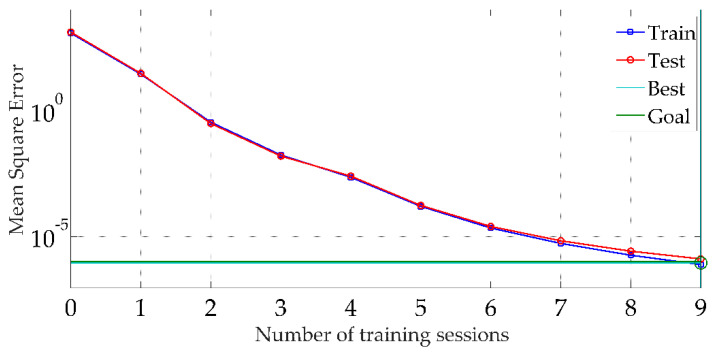
Mean Square Error curve of optimal neural network.

**Figure 7 sensors-22-04071-f007:**
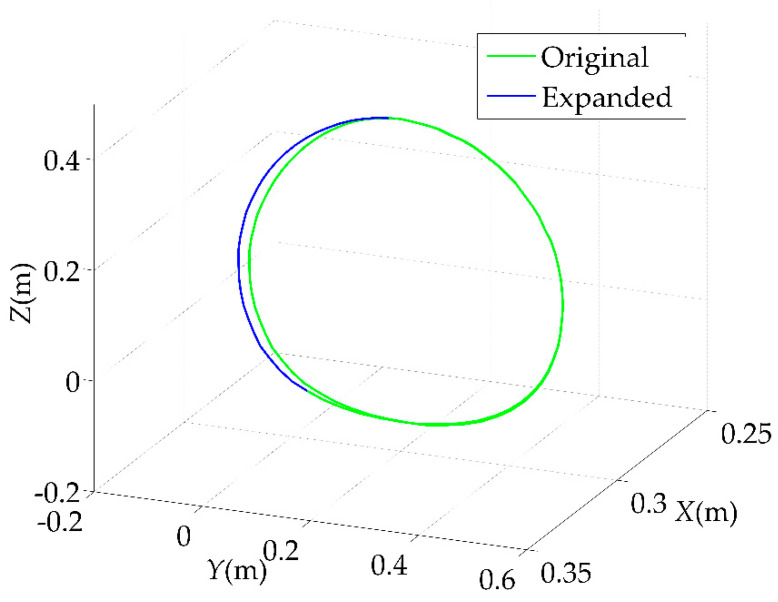
Rehabilitation trajectory under Cartesian space.

**Figure 8 sensors-22-04071-f008:**
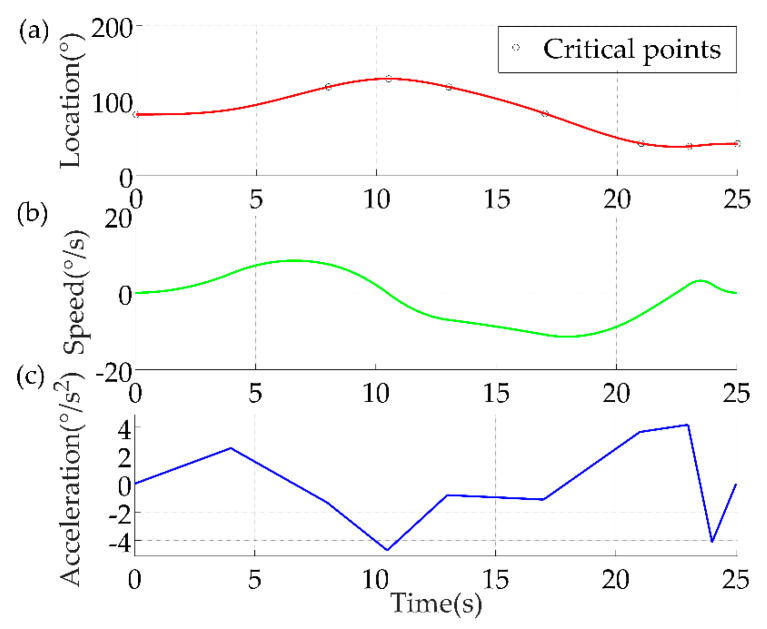
Results of kinematic analysis of joint #1. (**a**) Location; (**b**) Speed; (**c**) Acceleration.

**Figure 9 sensors-22-04071-f009:**
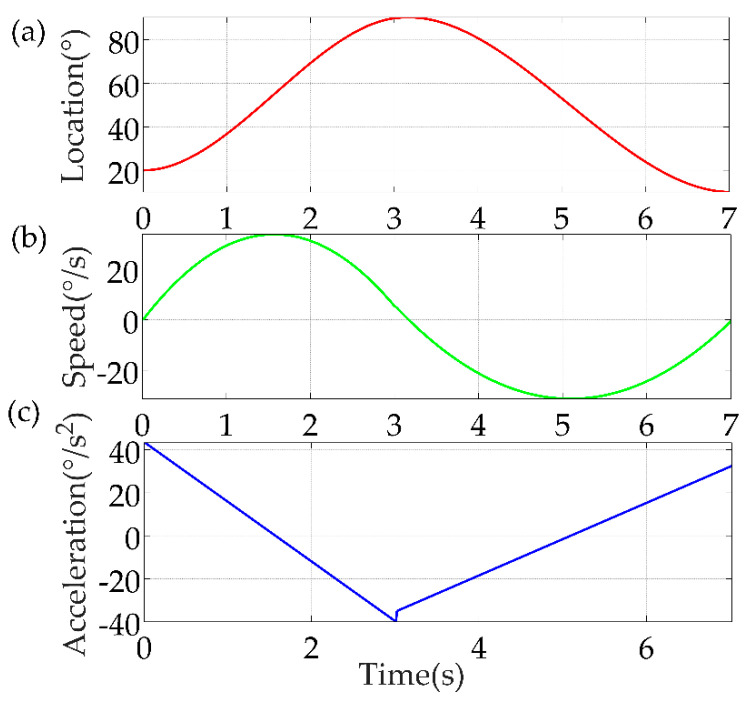
Results of MATLAB simulation of cubic polynomial interpolation method. (**a**) Location; (**b**) Speed; (**c**) Acceleration.

**Figure 10 sensors-22-04071-f010:**
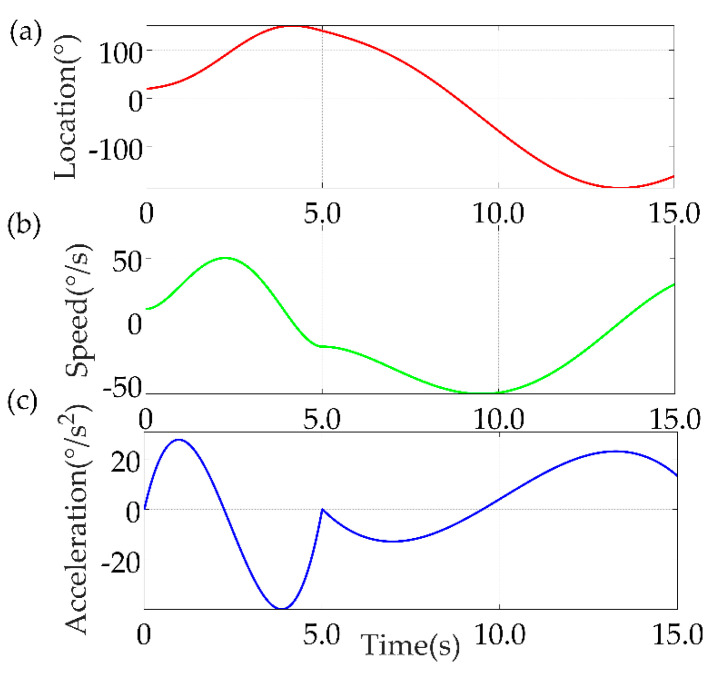
Results of MATLAB simulation of five−polynomial interpolation method. (**a**) Location; (**b**) Speed; (**c**) Acceleration.

**Figure 11 sensors-22-04071-f011:**
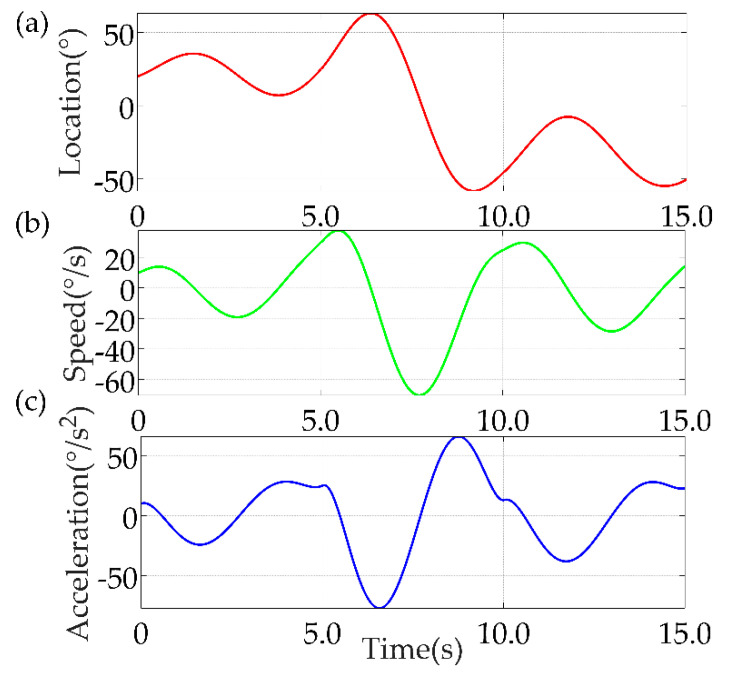
Results of MATLAB simulation of seven−polynomial interpolation method. (**a**) Location; (**b**) Speed; (**c**) Acceleration.

**Figure 12 sensors-22-04071-f012:**
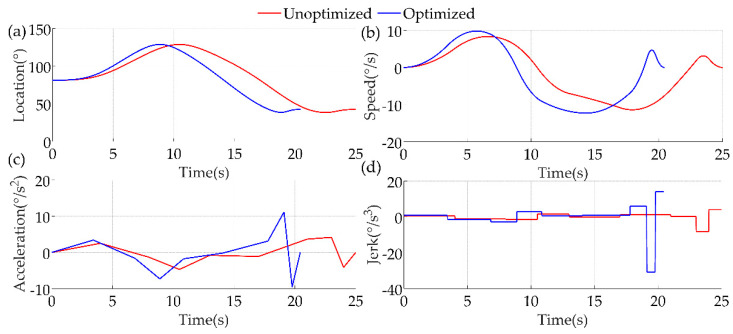
Comparison of joint #1 kinematic analysis before and after optimization. (**a**) Location; (**b**) Speed; (**c**) Acceleration; (**d**) Jerk.

**Figure 13 sensors-22-04071-f013:**
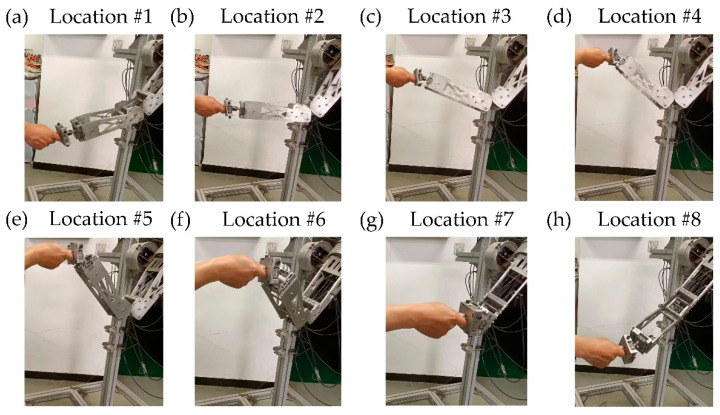
Position and posture of the treated person and the robotic arm during the rehabilitation process. (**a**) Location #1; (**b**) Location #2; (**c**) Location #3; (**d**) Location #4; (**e**) Location #5; (**f**) Location #6; (**g**) Location #7; (**h**) Location #8.

**Table 1 sensors-22-04071-t001:** Original radian data.

Group	Joint #1 (rad)	Joint #2 (rad)	Joint #3 (rad)	Joint #4 (rad)	Joint #5 (rad)	Joint #6 (rad)	Joint #7 (rad)
1	−1.4765	0.1257	0.0768	−1.8431	−1.4451	−0.5515	−0.5515
2	−1.4718	0.1319	0.0806	−1.8392	−1.4388	−0.5486	−0.5486
3	−1.4671	0.1382	0.0845	−1.8354	−1.4326	−0.5456	−0.5456
4	−1.4624	0.1445	0.0883	−1.8315	−1.4263	−0.5426	−0.5426
5	−1.4577	0.1508	0.0922	−1.8277	−1.4200	−0.5397	−0.5397
6	−1.4530	0.1571	0.0960	−1.8239	−1.4137	−0.5367	−0.5367
7	−1.4483	0.1634	0.0998	−1.8200	−1.4074	−0.5337	−0.5337
8	−1.4436	0.1696	0.1037	−1.8162	−1.4012	−0.5308	−0.5308
9	−1.4388	0.1759	0.1075	−1.8123	−1.3949	−0.5278	−0.5278
10	−1.4341	0.1822	0.1114	−1.8085	−1.3886	−0.5248	−0.5248

**Table 2 sensors-22-04071-t002:** Radian data solved by the trained network.

Group	Joint #1 (rad)	Joint #2 (rad)	Joint #3 (rad)	Joint #4 (rad)	Joint #5 (rad)	Joint #6 (rad)	Joint #7 (rad)
1	−1.4772	0.1253	0.0763	−1.8426	−1.4448	−0.5510	−0.5508
2	−1.4723	0.1314	0.0802	−1.8389	−1.4385	−0.5481	−0.5480
3	−1.4674	0.1376	0.0841	−1.8352	−1.4322	−0.5452	−0.5452
4	−1.4625	0.1438	0.0881	−1.8315	−1.4259	−0.5424	−0.5423
5	−1.4576	0.1500	0.0920	−1.8278	−1.4196	−0.5395	−0.5395
6	−1.4527	0.1561	0.0959	−1.8241	−1.4133	−0.5366	−0.5367
7	−1.4478	0.1632	0.0998	−1.8204	−1.4071	−0.5337	−0.5338
8	−1.4429	0.1685	0.1038	−1.8167	−1.4008	−0.5309	−0.5310
9	−1.4380	0.1747	0.1077	−1.8130	−1.3945	−0.5280	−0.5282
10	−1.4331	0.1809	0.1116	−1.8093	−1.3882	−0.5251	−0.5254

**Table 3 sensors-22-04071-t003:** Comparison of robotic arm inverse kinematics solution methods.

Method	Degree of Freedom	Maximum Error	Mean Square Error
This study	7DOF	0.002 rad	Approx. ^1^ 10^−6^
Jacobian transpose [29]	3DOF	Approx. ^1^ 27 mm	
Neural network [30]	Planar two and three−link manipulators	Approx. ^1^ 0.1 rad	Between 10^−2^ and 10^−3^
MPGA ^2^ [31]	6DOF	The error is of order 10^−2^	
FA ^3^ [32]	7DOF	6.538 × 10^−2^ mm	1.4547 × 10^−5^
ABC ^4^ [32]	7DOF	0.5475 mm	1.1105 × 10^−6^

^1^ Approximately. ^2^ Multiple Population Genetic Algorithm. ^3^ Firefly Algorithm. ^4^ Artificial Bee Colonies.

**Table 4 sensors-22-04071-t004:** Time comparison before and after track optimization.

Track Points	Angle (°)	Specify Time (s)	Optimized Time (s)
1~2	81.40~118.09	8.0	6.84
2~3	118.09~128.92	2.5	2.05
3~4	128.92~117.92	2.5	1.96
4~5	117.92~82.55	4.0	3.18
5~6	82.55~42.88	4.0	3.75
6~7	42.88~38.98	2.0	1.32
7~8	38.98~42.42	2.0	1.34

## Data Availability

Not applicable.

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
