# Peer review of "Trajectory Planning and Simulation Study of Redundant Robotic Arm for Upper Limb Rehabilitation Based on Back Propagation Neural Network and Genetic Algorithm"

_sensors, 2022, doi:10.3390/s22114071_

Round 1

Reviewer 1 Report

Dear Authors,

My comments and questions are listed in the attachment.

Kind Regards

Author Response

Dear Reviewer,

My responses are listed in the attachment.

Kind Regards

Reviewer 2 Report

I would like to express my gratitude regarding the opportunity to review this manuscript.

It is an interesting study on a relevant topic. At this early stage the manuscript requires improvements, described below with specific indication of the lines:

Title, authors, and affiliations – Please insert according to journal template.

Please consider improving the abstract aiming clear messages for readers understanding.

26 – Please consider reorganizing the introduction section. The paragraphs are too long and in many cases with references in full (with name and “et al”.).

82 – A reference is suggested for “Köker's idea”.

96 to 103 – The aim of the study should be better explained.

138 – Please consider avoiding “etc.”.

187-188 – Please consider line spacing.

214 – References end in page 6 and line 214. Please consider reorganizing the manuscript content, based on a regular reference to science throughout the pages. Currently the manuscript only presents 23 references, there is room for a more scientific nature in the manuscript.

242 – Please review the upper and lowercase in the figures and throughout all the manuscript, aiming standardization.

260 – Please carefully review tables 1 and 2 content and format.

273-277 – Please review text format.

323-326 – Please review text format.

328-332 – Please review text format.

406-408 – For example in these lines and in other parts of the manuscript, the text format is not according to the journal template. Please carefully review this.

431 – Please consider table format aiming the variables and units in the same line.

431 – Please review the decimals (81.4).

444 – Please consider improving figure quality.

446 – Please consider describing limitations or possible constraints regarding the scientific procedures.

447 – “Conclusions”. Please consider a clear take home message with suggestions for future studies.

English should be carefully reviewed throughout the manuscript.

References format should be carefully reviewed and corrected. They don’t seem according to the journal instructions.

Author Response

(The authors gave the same response as above.)

Reviewer 3 Report

Dear authors,

First of all, I would like to congratulate you for the very explicit work you have developed. I think you have explained in a very complete way the methodology you have followed in the design of your robot.

However, from a clinical point of view, I think that there are some unanswered questions:

  1. Title: the type of pathology to which this robot is addressed should be included.
  2. It is intuited from the introduction that they refer to the rehabilitation of an arm with hemiparesis as a consequence of a stroke. If so, why is the robotic arm you have designed appropriate for this pathology?
  3. How many patients have you tested your robot on? What pathology did they have?
  4. What was their level of satisfaction, once the robot had been tested?
  5. Do not include a discussion answering this question: What were your results in comparison to other similar robots?
  6. Conclusion: the conclusion should not be the summary of your work.

Best regards,

Author Response

(The authors gave the same response as above.)

Round 2

Reviewer 1 Report

Dear Authors,

You have adressed all my comments. The manuscript needs only minor revision (formet und spelling check). 

Kind Regards

Author Response

(The authors gave the same response as above.)

Reviewer 2 Report

The manuscript visibly improved after the authors' work, congratulations. Before suggestion for acceptance, there are still some structural issues regarding the manuscript that should be analyzed with the greatest care/detail. Below suggestions:

Title, authors, and affiliations – Please insert according to journal template (for example authors´ initials).

Please review the keywords standardization (upper and lowercase) according to the journal instructions for authors.

245 – “Finish” (suggested in uppercase like the others). BP and GA should be in full in the legend, aiming rapid interpretation by the readers.

All the figures do not seem according to the instructions for authors (type of letter Palatino Linotype, and other formats, size, etc.). Please carefully review.

295 – “Tables 1 and 2” is suggested.

303 – Please correct the tables line format (in particular the lower lines)

311 – MSE doesn’t need to appear abbreviated.

311 – Please consider formatting the table text and columns aiming text alignment.

504 – Table crosses two pages, please try in the same page aiming rapid interpretation by the readers.

520 – Figure 12 appears 2 times in the manuscript. Please confirm.

529-553 – Please consider reformulating the paragraph size, is too long.

594 – Please review authors contributions format.

606 – References format should be carefully reviewed and corrected. They don’t seem according to the journal instructions. Some examples: journal abbreviations, DOI, number and issue not separated by “,” and number in italic.

English should be carefully reviewed throughout the manuscript.

Author Response

(The authors gave the same response as above.)

Reviewer 3 Report

Dear authors, 
Thank you for your clarifications. 
From a clinical perspective, I continue to think that the information in the manuscript could be improved. 
Best regards, 

Author Response

(The authors gave the same response as above.)

Round 3

Reviewer 2 Report

The authors have done a praiseworthy job throughout the review process, congratulations. The suggestion is for accept after minor revision. A careful final analysis/reading with special emphasis on text details and manuscript formatting is suggested. Below are some examples of what could be improved:

298 – Please confirm if more than one space in the text (before “[31]”)

306 – Table 3 should present a legend for example with “APX”, “ABC” “FA” and “MPGA“descriptions.

532-559 – The size of the paragraphs is too different. Please consider reformulate aiming providing better reading conditions.

Please carefully review and correct the references according to the journal template and instructions for authors. For example, after de journal name sometimes “,”, others “.” and even nothing before the year of publication.

Author Response

Dear Reviewer,

My response are listed in the attachment.

Kind Regards
